

# Validation of a commercially available mobile application for velocity-based resistance training

Danielle Anne Trowell[1], Angus G. Carruthers Collins[1],
Ashlee M. Hendy[2], Eric James Drinkwater[1] and
Claire Kenneally-Dabrowski[1]

[1] Centre for Sport Research, Institute for Physical Activity and Nutrition, Deakin University, Burwood, Victoria, Australia
[2] Institute for Physical Activity and Nutrition, Deakin University, Waurn Ponds, Victoria, Australia

Corresponding author
Danielle Anne Trowell,
d.trowell@deakin.edu.au

## ABSTRACT

**Background:** Velocity-based training (VBT) is commonly used for programming and autoregulation of resistance training. Velocity may also be measured during resistance training to estimate one repetition maximum and monitor fatigue. This study quantifies the validity of Metric VBT, a mobile application that uses camera-vision for measuring barbell range of motion (RoM) and mean velocity during resistance exercises.

**Methods:** Twenty-four participants completed back squat and bench press repetitions across various loads. Five mobile devices were placed at varying angles (0, ±10, and ±20°) perpendicular to the participant. The validity of Metric VBT was assessed in comparison to Vicon motion analysis using precision and recall, Lin's concordance correlation coefficient, and Bland-Altman plots. Proportional bias was assessed using linear regression.

**Results:** Metric VBT accurately detected over 95% of repetitions. It showed moderate to substantial agreement with the Vicon system for measuring RoM in both exercises. The average Limits of Agreement (LoA) for RoM across all camera positions were −5.45 to 4.94 cm for squats and −5.80 to 3.55 cm for bench presses. Metric VBT exhibited poor to moderate agreement with the Vicon system for measuring mean velocity. The average LoA for mean velocity were 0.03 to 0.25 m/s for squats and −5.80 to 3.55 m/s for bench presses. A proportional bias was observed, with bias increasing as repetition velocity increased.

**Conclusions:** Metric VBT's wide LoA for measuring RoM and mean velocity highlights significant accuracy concerns, exceeding acceptable levels for practical use. However, for users prioritizing repetition counts over precise RoM or mean velocity data, the application can still provide useful information for monitoring workout volume.

## INTRODUCTION

Resistance training is a fundamental training tool for the development of muscular strength for general wellbeing (*Westcott, 2012*) and athletic performance (*Suchomel, Nimphius & Stone, 2016*). During resistance training, the external load lifted by an individual is typically programmed as a relative percentage of the individual's previously determined one repetition maximum (1RM) (*e.g.*, >80% 1RM for strength training) (*Kraemer & Ratamess, 2004*). Training volume (*i.e.*, sets and repetitions) is often programmed according to established recommendations based on an individual's training goals (*e.g.*, 3–6 sets of 1–6 repetitions for strength training) (*Bird, Tarpenning & Marino, 2005*; *Kraemer & Ratamess, 2004*; *Weakley et al., 2021a*). An individual's 1RM can fluctuate between individual training sessions because of factors such as athlete readiness, training-induced adaptions, or fatigue (*Greig et al., 2020*). There are also significant inter-individual differences in the volume of training that individuals can complete at specific relative loads (*Reynolds, Gordon & Robergs, 2006*; *Richens & Cleather, 2014*). Relying solely on an individual's previously measured 1RM and standard training volume guidelines to prescribe daily training volume may lead to individuals overtraining or undertraining. This approach can prevent an individual from effectively reaching their specific training goals (*Reynolds, Gordon & Robergs, 2006*; *Weakley et al., 2021a*).

Velocity-based training (VBT) has emerged as an alternative method for programming and autoregulating resistance training (*Larsen, Kristiansen & van den Tillaar, 2021*; *Weakley et al., 2021a*). The premise of VBT is based on the consistent and predictable relationship between relative external loads and lifting velocity (*Banyard, Nosaka & Haff, 2017*; *Weakley et al., 2021a*). This relationship shows that the velocity of a lift decreases as the external load increases, while lighter loads are associated with faster lifting velocities. For instance, during the full back squat, both mean velocity and mean propulsive velocity (MPV) have a strong relationship with %1RM, as evidenced by a high $R^2$ value of 0.96 (*Sánchez-Medina et al., 2017*). The relationship between relative load and velocity, however, varies in its linearity across different loads, exercises, and the specific velocity measure used (*García-Ramos et al., 2018*; *González-Badillo & Sánchez-Medina, 2010*). This predictable relationship allows individuals to regularly estimate their 1RM strength from their velocity when lifting submaximal loads, which can then be used to inform programming before each training session (*Weakley et al., 2021a*). Individuals can establish specific velocity targets (*e.g.*, 0.7–0.8 m/s) for their training sessions and adjust the load accordingly between sets to meet these goals. Furthermore, when each repetition is performed with maximal effort then neuromuscular fatigue will be accompanied by an involuntary decrease in movement velocity (*González-Badillo et al., 2017*; *Sanchez-Medina & González-Badillo, 2011*). This allows individuals to objectively monitor their fatigue by measuring the relative magnitude of velocity loss between each exercise set. The tolerated amount of velocity loss will depend on the specific exercise, training goal, and individual experience (*González-Badillo et al., 2017*).

Three-dimensional (3D) motion capture is well-recognized as the gold-standard instrument for measuring movement velocity (*Pérez-Castilla et al., 2019*; *Weakley et al.,*

*2021b*). Since 3D motion capture is not affordable or readily accessible for use during resistance training sessions, a mobile application named Metric VBT to measure barbell velocity using iOS and iPad OS devices. Metric VBT analyses mobile camera footage using a proprietary computer vision system and repetition-detection algorithm to report barbell range of motion (RoM) and mean velocity during resistance exercises. To date, the Metric VBT application has not been validated against a gold-standard criterion. It is imperative that a validation study be performed for the Metric VBT application as small differences in velocity can have critical implications for strength and conditioning practices.

For example, an increase in mean concentric velocity of 0.07–0.10 m/s is associated with a ~ 5% increase in 1RM strength in the full back squat (*Courel-Ibáñez et al., 2019*; *González-Badillo & Sánchez-Medina, 2010*; *Sánchez-Medina et al., 2017*). Therefore, the primary aim of this study is to quantify the validity of outcome measures obtained from the Metric VBT mobile application (repetition-detection, barbell RoM, mean barbell velocity) designed to monitor VBT in gym-based settings. To facilitate practical use, a secondary aim of this study is to understand how varying the physical set up of the camera device (*i.e.*, set up angle) within the manufacturers recommendations influences bias in outcomes measures. We hypothesized that i) Metric VBT will provide acceptable accuracy for repetition-detection, barbell RoM and mean barbell velocity, when compared to 3D motion capture, and ii) when the device is set up at larger angles (*i.e.*, further from perpendicular to the participant) the bias will increase.

## MATERIALS AND METHODS

### Experimental approach to the problem

We quantified the validity of the Metric VBT mobile application (version 0.6.0; Core Advantage Pty Ltd., Melbourne, Australia) by comparing its outputs with a Vicon 3D motion analysis system (Oxford Metrics Ltd, Oxford, UK). Participants completed repetitions of the barbell back squat at 25%, 50%, 70%, 90%, and 100% of 1RM, and the bench press at 50%, 70%, 90%, and 100% of 1RM. The squat and bench press exercises were chosen since they are fundamental compound lifts for the lower and upper body, respectively. Five mobile devices were placed at a range of angles (−20, −10, 0, 10, and 20°) to the right-hand side of the participant during the exercises (see Fig. S1, which shows a diagram of the data collection set up). All five mobile devices utilized version 0.6.0 of the Metric VBT application to simultaneously detect repetitions and measure barbell RoM and mean velocity by tracking a weight plate during the exercises. A Vicon motion analysis system was used to simultaneously track the 3D coordinates of markers placed on the ends of the barbell, which were used to measure the barbell RoM and mean velocity. The validity of the Metric VBT application at each of the five camera positions was assessed using precision and recall, as well as Lin's concordance correlation coefficient (CCC) and Bland-Altman plots. The presence of proportional bias at each camera position was assessed using linear regression.

## Participants

Twenty-four healthy participants (15 males and nine females) were recruited for this study. Participants were recruited from the local community and had a variety of resistance training experience ranging from novice (*e.g.*, twice in the past 3 months) to highly experienced (*e.g.*, 5–6 times per week for over 10 years). Eligibility criteria required that participants were over 18 years old and free from any current musculoskeletal injuries or conditions that may impact their ability to perform resistance training. Participants mean (SD) age, height, and mass were 27.0 (5.9) years, 1.79 (0.09) m, and 79.1 (15.3) kg, respectively. All participants were informed of the benefits and risks of this study before providing written informed consent in accordance with the ethical requirements of the Deakin University Human Research Ethics Committee (approval no: 2022-025).

## Equipment

Metric VBT is a mobile application available on iOS and iPad OS (*Metric Ltd., 2022*). Metric records videos in high-definition 720 p resolution from the rear or front-facing cameras at 60 frames-per-second. The mobile device must remain stationary and stable during filming. The manufacturers recommend positioning the mobile device within 0–25° of the end of the barbell, and at waist height or above when used for squats and the bench press. The capture space should be well-lit and the barbell, weight plates, and lifter should remain within the camera's field of view for the entire lift. Metric VBT measures barbell displacement by finding and tracking standard regulation weight plates of 450 mm (17 inch) diameter during the concentric phase of each repetition. Consequently, a weight plate of 450 mm diameter must be on the barbell and outer layer of a plate stack to enable video tracking. Barbell RoM (cm) is the distance the barbell moves vertically during the concentric phase of the movement. Mean barbell velocity (m/s) for each repetition is derived from the resultant (*i.e.*, combined vertical and anteroposterior axes) change in barbell displacement over change in time.

Data were collected in a well-lit indoor biomechanics laboratory. A power rack was placed in the middle of the capture space to allow participants to safely lift heavy weights. Two barbells (10 and 20 kg; Eleiko, Halmstad, Sweden), a variety of weight plates (1.5 to 25 kg; Eleiko, Halmstad, Sweden), and an adjustable weight bench were used during the testing session. We used five devices running iOS and iPadOS version 15.4.1 (Apple Inc., Cupertino, CA, USA) with Metric VBT installed. The devices included an iPhone 11 (model: MWLT2X/A), iPhone XR (model: A2105), iPhone 13 Pro (model: MNE23X/A), and twelfth generation iPad Pros (model: MY2H2X/A). All devices were equipped with 12 MP rear camera systems, configured to record in high definition at 720 p resolution and 60 frames-per-second. These settings were in accordance with the recommended device configurations by Metric VBT. Each device was mounted on a tripod positioned 1.50 m away from the right side of the power rack. The cameras were aimed to record participants from their right side, with the camera lenses centered at a height of 1.30 m from the ground. A total of five devices were placed at a range of angles (−20, −10, 0, 10, and 20°) to the right side of the power rack to assess validity of the application across the manufacturers recommended positioning guidelines (see Fig. S1, which shows a diagram

of the data collection set up). At the time of data collection, the Metric VBT application did not store results, so the display screens of all devices were filmed using a Panasonic HC-VC870M camcorder (Panasonic Holdings Corp., Osaka, Osaka, Japan) following each exercise set. Two-retroreflective markers (14 mm) were taped to the ends of the barbell and concurrent validity was assessed with 3D kinematic data from an eight-camera Vicon motion analysis system sampling at 200 Hz. The motion capture cameras were positioned on tripods surrounding the power rack and were 2.0 m from the rack. The Vicon motion capture system is known for its high-level accuracy and precision (*Eichelberger et al., 2016*; *Merriaux et al., 2017*). The system achieves a mean absolute error of just 0.15 mm in static conditions and less than 2 mm during dynamic movements (*Merriaux et al., 2017*). During walking, the Vicon system records marker positions with trueness and uncertainty levels of −0.08 and 0.33 mm, respectively (*Eichelberger et al., 2016*).

## Procedures

Participants attended the biomechanics laboratory wearing comfortable exercise clothing and shoes. Participants performed a standardized warm-up involving 10 min of stationary cycling followed by mobility exercises and submaximal barbell back squat and bench press exercises. 1RM was subsequently determined by progressively increasing the weight lifted during the squat and bench press exercises until participants were unable to increase the resistance further (*Triplett, 2016*). Participants were allowed 3 min of rest between 1RM attempts, and the test was ceased when the participant could not increase the resistance or could not maintain correct technique. The 1RM result was the heaviest load successfully lifted with correct technique. Following a self-selected rest period, participants completed five repetitions of the barbell back squat at loads corresponding to 25%, 50%, and 70% of 1RM, three repetitions at 90% of 1RM, and one repetition at 100% of 1RM (19 total repetitions). Participants then completed five repetitions of the bench press at loads corresponding to 50% and 70% of 1RM, three repetitions at 90% of 1RM, and one repetition at 100% of 1RM (14 total repetitions). The bench press at 25% of 1RM was excluded from analysis because this load was often lower than the minimum possible combined weight of the lightest barbell (15 kg) and weight plates with 450 mm diameter for video tracking (2.5 kg each). Each set was separated by 2–3 min of passive rest. Participants were instructed to perform the concentric phase of each movement as fast as possible.

## Data processing

The barbell RoM and mean velocity for each repetition and device were transcribed from the video recordings into a Microsoft Excel spreadsheet (Microsoft Corp., Redmond, WA, USA). Data from any additional repetitions identified by the Metric VBT application were also recorded in the spreadsheet. Raw marker data from the 3D motion capture system were reconstructed and labelled in Vicon Nexus software (version 2.12.1; Oxford Metrics Ltd, Oxford, UK). Vicon trials were manually cropped to avoid false repetitions (*e.g.*, racking and unracking) then labelled data were processed using custom MATLAB scripts (The Mathworks Inc., Natick, MA, USA). Marker trajectories were low-pass filtered using a zero-lag fourth-order Butterworth filter with a cut-off frequency of 10 Hz (based on a

residual analysis of data). The mid-point of the barbell was found by averaging the 3D spatial coordinates of the right and left barbell markers. Local minima and maxima points in the vertical position of the mid-barbell were used to define the start and end times of the concentric phase of each repetition, respectively. Only the concentric phase was analyzed for both exercises. Barbell RoM (cm) was calculated as the change in vertical position of the barbell mid-point from the start (*i.e.*, bottom) to the end (*i.e.*, top) of each repetition. Mean barbell velocity (m/s) was calculated as the resultant (*i.e.*, combined vertical and anteroposterior axes) displacement of the barbell mid-point divided by the change in time using the previously determined start and end times for each repetition. The start and end times used in the mean barbell velocity calculation were derived from the vertical position of the barbell mid-point to remain consistent with the Metric VBT algorithm. Barbell RoM and mean velocity data were exported from MATLAB to the Excel spreadsheet where output data for each repetition from the Metric VBT application and Vicon motion system were matched.

## Statistical analyses

The number of true positives (TP; a repetition detected on Metric VBT and Vicon), false positives (FP; a repetition detected on Metric VBT but not by Vicon) and false negatives (FN; a repetition detected by Vicon but not by Metric VBT) were computed (*Kelly et al., 2012*). This allowed calculation of precision (the ability of Metric VBT to correctly identify a repetition, *i.e.*, low FP and high TP) and recall (the ability of Metric VBT to correctly detect a repetition, *i.e.*, low FN and high TP).

The difference in barbell RoM and mean velocity between the Metric VBT application and Vicon were calculated for each repetition (*i.e.*, Metric VBT–Vicon). The difference data were then imported into SPSS (version 29.0.0; IBM Statistics, Chicago, IL, USA) for statistical analyses. Visual inspection of histograms and box plots indicated that the difference data were approximately normally distributed. Lin's CCC along with a 95% confidence interval (CI), were used to measure the agreement between conditions (*Lawrence & Lin, 1989*). The CCC values were classified as almost perfect (>0.99), substantial (0.95–0.99), moderate (0.90–0.95), and poor (<0.90) according to *McBride (2005)*. Bland-Altman analyses were conducted to calculate and visually represent the bias at each of the five camera positions for each exercise (*Bland & Altman, 1999*). Linear regressions were used to check for systematic or proportional bias in the Metric VBT results. Proportional bias was identified if the slope of the regression line was significant ($p < 0.05$). Proportional bias was then identified as monophasic or biphasic (*Ho, 2018*). Monophasic bias skewed data in a consistent direction, either overestimating or underestimating, with bias magnitude increasing proportionally with the measurement. In contrast, biphasic bias changed direction at a specific threshold (*e.g.*, it underestimated values below a specific threshold and overestimated above it) with bias still proportional to the measurement size.

## RESULTS

1RM were 96.81 ± 35.74 kg and 64.17 ± 32.23 kg for the squat and bench press exercises, respectively. The average barbell RoM during the squat was 56.00 ± 9.47 cm while the average RoM during the bench press was 42.89 ± 6.32 cm (as measured by Vicon). A total of 455 back squat repetitions and 330 bench press repetitions were analyzed across varying loads. The Metric VBT application was not utilized at the −20° camera position for 75 squat repetitions and 46 bench press repetitions (due to miscellaneous device or application issues), and these repetitions were excluded from the analysis for this camera position.

### Precision and recall

The precision and recall of the Metric VBT application for the back squat and bench press exercises are reported in Table 1. During the squat, the small number of false negatives most often occurred at light loads and high velocities (*i.e.*, $n = 3$ at 25% repetition maximum (RM); $n = 1$ at 100% RM). During the bench press, false negatives were most common at high loads and low velocities (*i.e.*, $n = 2$ at 70% RM; $n = 13$ at 90% RM; $n = 11$ at 100% RM). False positives were observed across all loads during the bench press ($n = 17$ at 50% RM; $n = 19$ at 70% RM; $n = 15$ at 90% RM; $n = 20$ at 100% RM).

### Agreement: back squat

Moderate agreement was achieved when comparing RoM measured by the Metric VBT application at the −10° and −20° camera positions to Vicon (CCC: 0.949 and 0.941, respectively). Substantial agreement was achieved at the 0°, +10°, and +20° camera positions (CCC: 0.958–0.975). The mean bias was small (less than 1 cm) when comparing RoM measured by the Metric VBT application to Vicon; however, the lower and upper Limits of Agreement (LoA) were relatively large (−5.45–4.94 cm on average; Table 2). Mean bias was smallest at the 0° camera position, and the magnitude of error increased at camera positions further from perpendicular to the participant. Biphasic proportional bias was identified at the 0°, +10°, and +20° camera positions. This reflected the tendency for the Metric VBT application to overestimate RoM at small RoMs and underestimate RoM at large RoMs (Fig. 1A).

Poor agreement was achieved when comparing mean velocity measured by the Metric VBT application at all camera positions to Vicon (CCC: 0.826–0.890). The Metric VBT application overestimated mean repetition velocity at all camera positions (mean bias = 0.09–0.13 m/s; Table 2) when compared to Vicon. The lower and upper LoA were, on average, −0.03–0.25 m/s. Monophasic proportional bias was evident, whereby the bias increased as the repetition velocity increased (*i.e.*, at lower loads; Fig. 1B).

The results for the 0° camera position are displayed in Fig. 1. Bland-Altman plots for other camera positions are presented in Data S2.

### Agreement: bench press

Moderate agreement was achieved when comparing RoM measured by the Metric VBT application at all camera positions to Vicon (CCC: 0.901–0.926). The Metric VBT

**Table 1 Comparison of the number of repetitions detected by Metric And Vicon.**

| Camera position | True positives | False positives | False negatives | Precision | Recall |
|---|---|---|---|---|---|
| **Squat** | | | | | |
| −20° | 380 | 0 | 0 | 1.000 | 1.000 |
| −10° | 454 | 0 | 1 | 1.000 | 0.998 |
| 0° | 455 | 0 | 0 | 1.000 | 1.000 |
| +10° | 453 | 0 | 2 | 1.000 | 0.996 |
| +20° | 453 | 0 | 2 | 1.000 | 0.996 |
| **Bench press** | | | | | |
| −20° | 281 | 18 | 3 | 0.940 | 0.989 |
| −10° | 325 | 19 | 5 | 0.945 | 0.985 |
| 0° | 325 | 14 | 5 | 0.959 | 0.985 |
| +10° | 323 | 11 | 7 | 0.967 | 0.979 |
| +20° | 324 | 9 | 6 | 0.973 | 0.982 |

**Table 2 Bland-Altman results for the squat exercise. Mean bias, limits of agreement (LoA), and proportional bias are presented for barbell range of motion (RoM) and mean velocity.**

| Camera position | Mean bias (cm) | Lower LoA (cm) | Upper LoA (cm) | Slope significance ($p$) | Regression equation | Monophasic/biphasic |
|---|---|---|---|---|---|---|
| **RoM** | | | | | | |
| −20° | −0.84 | −7.47 | 5.80 | 0.063 | – | – |
| −10° | 0.28 | −5.55 | 6.11 | 0.07 | – | – |
| 0° | −0.08 | −4.17 | 4.01 | <0.001[*] | y = −0.045x + 2.441 | Biphasic |
| +10° | −0.15 | −5.37 | 5.07 | <0.001[*] | y = −0.048x + 2.542 | Biphasic |
| +20° | −0.49 | −4.70 | 3.72 | <0.001[*] | y = −0.039x + 1.690 | Biphasic |
| **Velocity** | | | | | | |
| −20° | 0.09 | −0.04 | 0.22 | <0.001[*] | y = 0.048x + 0.057 | Monophasic |
| −10° | 0.13 | −0.03 | 0.28 | <0.001[*] | y = 0.126x + 0.027 | Monophasic |
| 0° | 0.10 | −0.01 | 0.21 | <0.001[*] | y = 0.068x + 0.049 | Monophasic |
| +10° | 0.11 | −0.05 | 0.28 | <0.001[*] | y = 0.109x + 0.030 | Monophasic |
| +20° | 0.10 | −0.04 | 0.24 | <0.001[*] | y = 0.076x + 0.039 | Monophasic |

**Note:**
[*] Significant slope (gradient of the regression line, indicating the presence of proportional bias).

application underestimated RoM at all camera positions (mean bias = −0.74 to −1.44 cm; Table 3) when compared to Vicon. The lower and upper LoA were, on average, −5.80–3.55 cm. Monophasic proportional bias was observed at the −10° camera position, where the bias decreased as the RoM increased. Biphasic proportional bias was identified at the 0° and +10° camera angles. This bias showed that the Metric VBT application generally underestimated the RoM, except when the true RoM was very large, at which point Metric VBT overestimated the RoM.

Poor agreement was achieved when comparing mean velocity measured by the Metric VBT application at the −10° and +10° camera positions to Vicon (CCC: 0.883 and 0.881, respectively). Moderate agreement was achieved at the 0°, −20°, and +20° camera positions

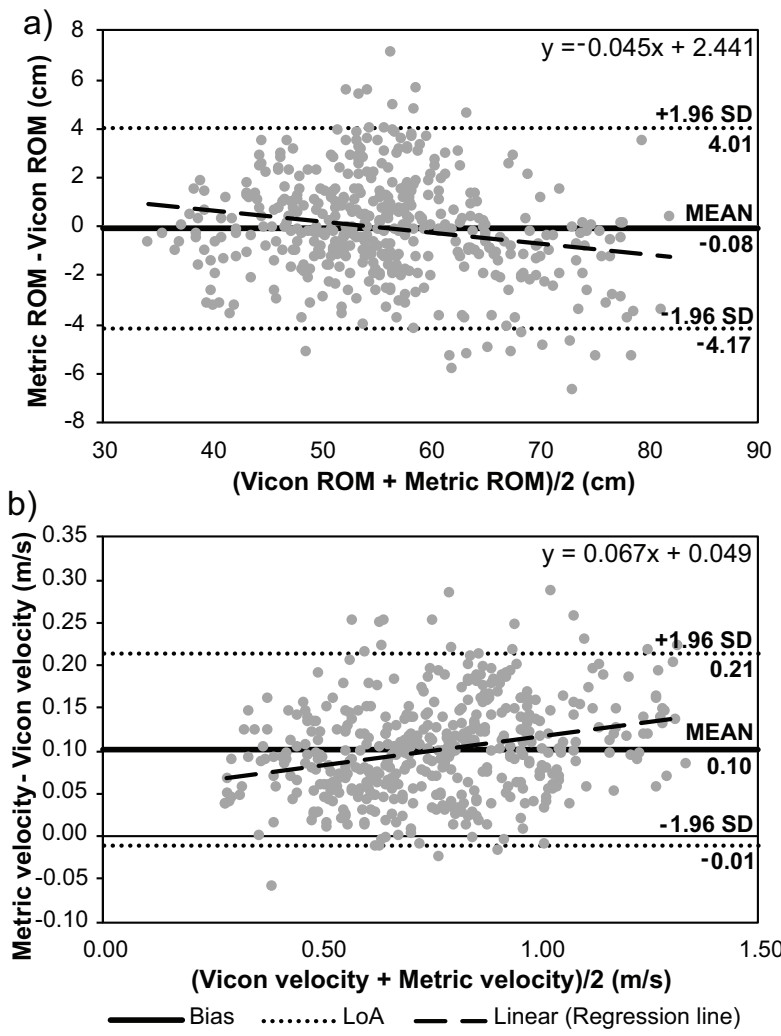

**Figure 1 Bland-Altman analyses displaying the mean bias and limits of agreement (LoA) between the Metric VBT application and Vicon motion analysis system during the back squat exercise.** (A) Barbell range of motion (RoM) and (B) mean barbell velocity for each repetition.

(CCC: 0.914–0.931). The Metric VBT application overestimated mean repetition velocity at all camera positions (mean bias = 0.06–0.09 m/s; Table 3) when compared to Vicon. The lower and upper LoA were, on average, −5.80–3.55 m/s. Monophasic proportional bias was observed at all camera positions, with greater bias observed at higher velocities (Fig. 2B).

The results for the 0° camera position are displayed in Fig. 2. Bland-Altman plots for other camera positions are presented in Data S2.

## DISCUSSION

This research quantifies the validity of Metric VBT (version 0.6.0), a mobile application that uses camera-vision for repetition-detection and measuring barbell RoM and mean velocity across a range of intensities during resistance exercises. Metric VBT was able to

**Table 3 Bland-Altman results for the bench press exercise. Mean bias, limits of agreement (LoA), and proportional bias are presented for barbell range of motion (RoM) and mean velocity.**

| Camera position | Mean bias (cm) | Lower LoA (cm) | Upper LoA (cm) | Slope significance (p) | Regression equation | Monophasic/biphasic |
|---|---|---|---|---|---|---|
| **RoM** | | | | | | |
| −20° | −1.44 | −6.08 | 3.20 | 0.167 | – | – |
| −10° | −0.96 | −5.82 | 3.90 | 0.026* | y = 0.050x − 3.092 | Monophasic |
| 0° | −1.15 | −6.04 | 3.74 | <0.001* | y = 0.073x − 4.267 | Biphasic |
| +10° | −0.83 | −5.85 | 4.19 | 0.031* | y = 0.052x − 3.042 | Biphasic |
| +20° | −0.74 | −5.19 | 3.71 | 0.287 | – | – |
| **Velocity** | | | | | | |
| −20° | 0.06 | −0.07 | 0.20 | <0.001* | y = 0.65x + 0.021 | Monophasic |
| −10° | 0.09 | −0.07 | 0.26 | <0.001* | y = 0.135x + 0.004 | Monophasic |
| 0° | 0.08 | −0.06 | 0.22 | <0.001* | y = 0.090x + 0.018 | Monophasic |
| +10° | 0.09 | −0.08 | 0.26 | <0.001* | y = 0.122x + 0.010 | Monophasic |
| +20° | 0.07 | −0.05 | 0.20 | <0.001* | y = 0.071x + 0.029 | Monophasic |

**Note:**
* Significant slope (gradient of the regression line, indicating the presence of proportional bias).

accurately detect over 95% of repetitions during the squat and bench press exercises. While the application demonstrated low mean bias for measuring RoM during the squat and bench press, the distribution of errors was highly variable (*i.e.*, relatively large LoA) and the application consistently overestimated mean barbell velocity during both exercises. Consequently, we partially accept our first hypothesis. Supporting our second hypothesis, the mean bias and LoA typically increased when cameras were positioned away from the optimal camera position (0°), indicating decreased accuracy as the camera moved further from being perpendicular to the participant.

The perfect precision and low occurrence of false negatives (<1%) across all camera positions indicate that the repetition recognition algorithm is reliable for the squat exercise. However, Metric VBT was less accurate at identifying bench press repetitions. The applications varying precision and recall at different camera angles during the bench press shows inconsistency and reduced practicality, with the highest precision observed at the +20° camera angle and optimal recall at 0° and −10°. This demonstrates the sensitivity of Metric VBTs repetition detection algorithm to changes in exercise types and camera perspectives. Nearly all false negatives (repetitions missed by Metric VBT) occurred during the 90% or 100% RM bench press trials when movement velocity is slowest. Metric VBT uses a 0.1 m/s velocity threshold to identify the start and end of each repetition. As a result, very slow repetitions at high loads are not detected, as observed in the current study. False positives were detected by Metric VBT during the bench press across all loads. These extra repetitions, averaging +4.5% across all camera positions, are likely a result of unracking the barbell and are therefore not a true repetition. Metric VBT has since added the capability for users to remove extra repetitions caused by unracking, but the burden is placed on the user to correctly identify these repetitions following each set. To minimize the detection of false repetitions, the Metric VBT algorithm also includes a condition stipulating that the starting point of each repetition must be below 40% of the global maximum (*i.e.*, the

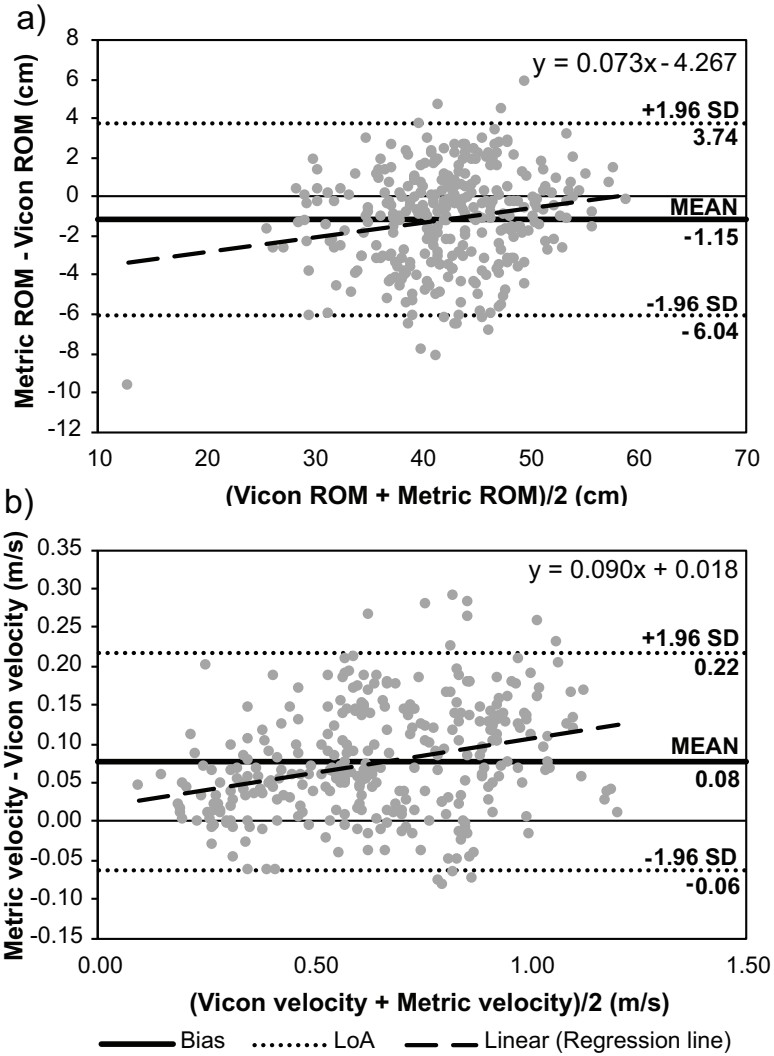

**Figure 2** **Bland-Altman analyses displaying the mean bias and limits of agreement (LoA) between the Metric VBT application and Vicon motion analysis system during the bench press exercise.** (A) Barbell range of motion (RoM) and (B) mean barbell velocity for each repetition.

highest vertical point recorded in each set) and the endpoint above 60% of the global maximum. Consequently, repetitions with a slow velocity (<0.1 m/s) during the initial phase of movement may show a reduced vertical range of displacement and fail to meet this criterion for repetition detection. Given this study's findings, the application's current repetition-detection algorithm should be adjusted to improve its precision and recall of repetitions during the bench press exercise.

The evaluation of the Metric VBT application for measuring RoM during the back squat identified limitations that compromise its suitability for this purpose. Despite achieving moderate to substantial agreement with the Vicon system across various camera angles, the wide LoA–averaging up to 10.4 cm–reveal issues with the application's accuracy. These wide LoA can lead to inaccurate measures of squat depth, which is an important metric for

minimizing excessive strain among clinical populations (*Zulkifley, Mohamed & Zulkifley, 2019*) and when monitoring and adjusting for fatigue-induced alterations in exercise form. During the bench press, the Metric VBT application showed only moderate agreement with the Vicon system. It consistently underestimated RoM with an average LoA of 9.5 cm across all camera positions, indicating significant issues with the application's ability to measure RoM accurately. Given these inconsistencies and their potential effects on training effectiveness and safety, the Metric VBT application is unsuitable for accurate RoM measurement in resistance training contexts.

The Metric VBT application demonstrated inaccuracies in mean velocity measured at all camera positions for both squat and bench press exercises, potentially leading to significant errors during VBT. There was poor agreement between Metric VBT and the Vicon system across all camera angles during the squat, with average LoA for mean velocity ranging 0.28 m/s. Considering that a change of 0.07–0.10 m/s in mean concentric velocity represents an approximate 5% change in 1RM strength for back squats (*Courel-Ibáñez et al., 2019*; *González-Badillo & Sánchez-Medina, 2010*; *Sánchez-Medina et al., 2017*), this variability suggests that for the same exercise intensity, the device may identify the relative effort inconsistently by ±25%. These errors in measuring exercise intensity compromise the effectiveness of VBT, which relies on the assumption that lift velocity is consistently predictable for given percentages of an individual's 1RM (*González-Badillo & Sánchez-Medina, 2010*). This is essential for accurately managing training intensity and adjusting loads to maintain the desired exercise intensity during VBT. Researchers (*Vernon, Joyce & Banyard, 2020*) found that decreases in squat velocity at 24- and 48-h post-strength training suggest incomplete recovery and reduced readiness to train. Mean velocity decreased up to 0.04 m/s across loads from 60% to 100% 1RM. These reductions are considerably smaller than the wide LoA observed with the Metric VBT application, indicating that Metric VBT also lacks the accuracy to reliably assess changes in velocity to determine training readiness. The LoA at all camera positions are also larger than the LoA previously reported for an alternative mobile application (*i.e.*, MyLift) during the back squat at loads of 40–100% of 1RM (0.01 ± 0.05 m/s) (*Thompson et al., 2020*). When measuring mean velocity during the bench press, Metric VBT had poor to moderate agreement with the Vicon system across camera positions. The broad LoA ranged up to 0.29 m/s and exceed the typical variation in mean velocity (−0.18 m/s) observed when performing the bench press to failure (*Duffey & Challis, 2007*). Thus, the Metric VBT system is not valid for detecting any meaningful changes in mean velocity because its measurement error is greater than the actual variation observed during the bench press.

This study used 3D motion analysis to measure mean barbell resultant velocity to determine whether Metric VBT accurately measures mean barbell resultant velocity as it purports to. Mean barbell velocity is calculated by the mobile application as the resultant (*i.e.*, combined vertical and anteroposterior axes) displacement of the barbell divided by the change in time. The change in time is determined from the start and end times described previously. That is, the repetition starts when the vertical velocity exceeds 1.0 m/s and the barbell is below 40% of the global maximum in the vertical axis, and the repetition stops when the vertical velocity slows below 0.1 m/s and the barbell is above 60%

of the global maximum in the vertical axis. The manufacturer's choice to report vertical RoM and mean resultant barbell velocity instead of mean vertical barbell velocity introduces inconsistency. Additionally, it is problematic that the application determines start and stop timepoints based on vertical position and velocity thresholds for calculating resultant velocity. These decisions complicate data interpretation and compromise accuracy, thereby diminishing the application's utility for VBT. Moreover, the fact that Metric VBT reports the mean velocity of the barbell rather than MPV further complicates its effectiveness for VBT. MPV specifically measures the velocity during the propulsive phase of a movement, excluding the braking phase where the movement slows down (*Sanchez-Medina, Perez & Gonzalez-Badillo, 2010*). This phase represents the portion of the movement when the muscles exert force to accelerate the resistance load. Consequently, it is commonly defined as a period of positive acceleration, where the force applied to the resistance load is greater than the gravitational pull (*i.e.*, $-9.81$ m/s$^2$), resulting in an increase in velocity (*Sanchez-Medina, Perez & Gonzalez-Badillo, 2010*). In contrast, mean velocity includes both the acceleration and braking phases of a movement. The inclusion of a significant braking phase in the calculation of mean velocity can obscure the relationship between load and velocity that is fundamental for VBT training (*Sanchez-Medina, Perez & Gonzalez-Badillo, 2010*). While Metric VBT uses a velocity threshold of 0.1 m/s when identifying the start and stop of each repetition, this may not accurately represent the true initiation or cessation of muscular effort. This could result in the inclusion of non-propulsive phases where the load is not actively being accelerated, thus distorting the mean velocity measurement. If the braking phase is long or particularly pronounced, such as with heavy loads or during fatigue, it can substantially lower the mean velocity making it appear as though the athlete is not generating as much power as they are during the propulsive phase of the exercise. Similarly, setting the velocity threshold too high might lead to the exclusion of valid data where the velocity is slightly less than 0.1 m/s but still represents significant muscular effort. Since MPV offers a more accurate reflection of performance than mean velocity, particularly at moderate or lighter exercise intensities (*Sanchez-Medina, Perez & Gonzalez-Badillo, 2010*), it is vital for any VBT tool to include this capability.

It is important to consider the limitations of this validation study when interpreting these findings. We used four different variations of mobile device with the Metric VBT application installed in this study. While all devices had the same camera and iOS settings, there may still be variations in how motion tracking algorithms perform across different devices due to variations in camera technology, processing power, and additional sensor capabilities between devices. All mobile devices were positioned at the same camera angle throughout data collection, which means variations in motion tracking algorithms could have influenced our comparisons between different camera angle positions. Nevertheless, given the poor performance of Metric VBT at various set up angles within the manufacturer guidelines, it is unlikely that any variations in mobile devices would alter the overall conclusions of this study. Finally, this study examined version 0.6.0 of the Metric VBT application. The findings from this validation study may not necessarily apply to later

versions of the Metric VBT application if the algorithms for repetition detection or measuring barbell RoM and mean velocity have been modified.

## CONCLUSIONS

Metric VBT showed acceptable precision and recall for detecting repetitions during squats; however, its accuracy was lower when identifying bench press repetitions. While it achieved moderate to substantial agreement with the Vicon system in measuring the RoM during both exercises, the wide LoA highlighted significant accuracy concerns. Moreover, both exercises displayed poor to moderate agreement with the Vicon system in measuring mean velocity, with LoAs that exceeded acceptable levels for practical applications. Therefore, while Metric VBT represents a promising technology for monitoring exercise performance, these findings indicate that it is currently insufficient for guiding VBT. Nonetheless, for users primarily interested in tracking repetition counts without the need for accurate RoM or mean velocity measurements, the application could still offer valuable data to help monitor workout volume.

## ACKNOWLEDGEMENTS

The authors would like to acknowledge the participants who took part in this study on a voluntary basis.

### Funding

The authors received no funding for this work.

### Competing Interests

The authors wish to declare a non-financial, academic, and professional relationship exists between Deakin University and Core Advantage, developers of the Metric application. Specifically, a small number of Deakin University undergraduate students are competitively selected annually to undertake professional placement hours on a voluntary basis at the Core Advantage Athletic training facility in Oakleigh. This relationship did not impact the outcomes of this research in any way.

### Author Contributions

- Danielle Anne Trowell conceived and designed the experiments, performed the experiments, analyzed the data, authored or reviewed drafts of the article, and approved the final draft.
- Angus G. Carruthers Collins conceived and designed the experiments, performed the experiments, analyzed the data, authored or reviewed drafts of the article, and approved the final draft.
- Ashlee M. Hendy conceived and designed the experiments, authored or reviewed drafts of the article, and approved the final draft.
- Eric James Drinkwater conceived and designed the experiments, authored or reviewed drafts of the article, and approved the final draft.

- Claire Kenneally-Dabrowski conceived and designed the experiments, performed the experiments, analyzed the data, prepared figures and/or tables, authored or reviewed drafts of the article, and approved the final draft.

## Human Ethics

The following information was supplied relating to ethical approvals (*i.e.*, approving body and any reference numbers):

Deakin University Human Research Ethics Committee.

## Data Availability

The individual data used within this study and the code used to process the data are available in the Supplemental Files.

## Supplemental Information

Supplemental information for this article can be found online at http://dx.doi.org/10.7717/peerj.17789#supplemental-information.

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
