# Peer review of "Validation of a commercially available mobile application for velocity-based resistance training"

_PeerJ, doi:10.7717/peerj.17789_

## Round 0.1 · original submission · Major Revisions

Please review all comments from reviewers. There was a diversity in opinions on the manuscript.

·

Basic reporting

Original title:
Validation of a commercially available mobile application for velocity-based resistance training
Review

Dear authors and editor, I am very happy to have the opportunity to help you rise the quality of this paper. It is already very high, which I really respect and apreciate, the paper is interesting and have the high practical value.
Please see my remarks below and make necessarry corrections:

Experimental design

73 please specify which 3d motion capture system was compared
76 add citation (who made this comparison?)
94 please consider changing to ‘ we hypothesized ‘
Material and methods:
Please specify the company and model of the mobile devices, were they all the same model and iOS version?
Please put the photo of the experimental setup if possible
Please add information which will clearly inform the readers that the monitoring was simoultanousely performer by 5 mobile devices of the same type and the same version of application and the vicon system all together – please be very informative to to readers
Please specify the Surface and direction in which the mobile devices were instlled in relations to the user
Please specify the load in kg (range and mean)
Please characterize the breaks between each tries of different % of 1RM – was this regulated or self-dependend for each participant?
Please specify the method of retriveing the 1RM value – the number of tries and time of the brakes between each try. Please add info about the 1RM range
113 corect – probably to ‘twice a week in the past 3 months’
121 please specify the time of being free from injuries because being free from injuries from the whole life is of low possibility in sport and even in recreation or real life conditions
Please inform if the dropping the bar at the chest was allowed to the participants during the bench press?
Please inform if the motion in the down or up direction was montiored or both in two analysed exercieses
Agreement: please add in 246-247, 249-250 please add ‘when comparing to Vicon 3d MC system” in all comparisons

Validity of the findings

the impact and novelty is of high level.
the data files and tables are clear

Additional comments

300 and 302 measured by what method and equpiment?

Reviewer 2 ·

Basic reporting

All comments are provided in the "Additional comments section".

Experimental design

All comments are provided in the "Additional comments section".

Validity of the findings

All comments are provided in the "Additional comments section".

Additional comments

This study tried to answer whether Metric VBT can be a useful tool for training monitoring. Generally, the introduction and method sections are well-written, while the discussion and conclusion sections need in-depth reorganization. Please see the comments below.
Line 39: one repetition maximum should be abbreviated 1RM, not RM.
Can you please rewrite the following sentence: “Relying on an individual’s previous 1RM and general recommendations for prescribing training volume can contribute to individuals training incorrectly for their intended goals (Reynolds et al. 2006; Weakley et al. 2021a).”?
Suppose the authors decided to use the three-dimensional system to validate the Metric VBT application. In that case, I think there is no need to write so much about linear position transducers and accelerometers (Lines 67-79). It would be more appropriate to only mention these two device types, and justify the creation of the Metric VBT as a more affordable solution for monitoring resistance training velocity.
Line 220: Isn´t the “consistent over- or under-estimation throughout the range of velocities measured” systematic bias, rather than monophasic proportional bias?, while biphasic bias “(the tendency to over- or under-estimate changes as a function of the velocity measured)” proportional bias? Please explain.
Line 261: change hyphen to “to” better: (here: -0.74 - - 262 1.44 cm)
This sentence is a bit confusing: Biphasic proportional bias was identified at the 0° and +10° camera positions, which reflected a tendency for the Metric VBT application to underestimate RoM at most RoMs while overestimating very large RoMs.
The second paragraph of the discussion should contain fewer results (numbers) and more discussion.
Line 300: there is an issue with the reference Duffey et al.
Line 333: same here
Line 344: same here and many more, please check
Third, fourth, and fifth, which delve into velocity-based training, can be condensed into a single paragraph. These points underscore the inadequacy of Metric VBT in accurately measuring velocity output. As such, discussing all potential applications of VBT becomes unnecessary, as the primary objective relies on obtaining precise velocity data for effective implementation.
Lines 395-399: It is not acceptable to leave readers with many doubts in the discussion section.
The whole discussion needs to be rewritten because it contains many numbers (result repetition) and it is unnecessarily long. I suggest implementing the following paragraphs into the discussion:
1. Main findings summarized
2. Discussion regarding the detection of the repetitions and position of the camera
3. Discussion regarding range of motion
4. Discussion regarding mean velocity
5. Limitation section
I think that the conclusions should be completely rewritten. Upon reviewing the data and results section, it is apparent that Metric VBT's efficacy in measuring mean velocity is questionable, warranting its discouragement for such applications. Its sole practicality lies in detecting repetitions. However, this limited functionality falls short of rendering Metric VBT a viable tool for comprehensive training monitoring. This should be the main idea of the conclusion section.
There should be no numbers in the conclusion section.

Reviewer 3 ·

Basic reporting

Validation of a commercially available mobile application for velocity-based resistance training (#94383)

This research aims to quantify the validity of Metric VBT. This mobile application uses camera vision for repetition detection, measuring barbell RoM and mean velocity across various intensities during resistance exercises. This validation attempt was carried out by comparing the output of Metric VBT with a Vicon 3D motion analysis system. The results indicate that the new measuring instrument has significant errors in the measurement of distances and mean velocity.

General and specific comments

The study's results lead us to the conclusion that this new instrument should not be used to control execution velocity. In this regard, trainers should not be advised to "consider the application's sensitivity when using the device", but rather to discourage its use. Furthermore, we believe that it is pertinent to consider the study's justification and methodology.

The study applies a small number of tests to check the agreement of the new device with the reference device. However, the differences between the two devices are so significant that, in this case, it would not even be necessary to apply further tests to ratify the results.
However, a study of these characteristics should have included the application of other tests such as, for example, Lin's correlation coefficient of concordance (CCC_Lin), the regression line between the measurements of the two devices, assessing their proximity to the bisector of the angle of the coordinate axes, not only the value of the correlation, the value of the mean of the differences, which indicates the systematic error, as well as the variance of the differences between the measurements, which would indicate the dispersion of the random error of the variance or imprecision of the measurements.
A possible relevant problem of the study is the choice of the reference device. Although it is considered a validated device for measuring displacement distance and velocity, no evidence is presented that it has been validated for this type of measurement.

The study records mean velocity without mentioning mean propulsive velocity (MPV). The authors should know that MPV is a more accurate performance indicator than mean velocity when the relative intensity is moderate or light. Studies on the origin of VBT (some of them cited in the manuscript) always refer to MPV since its measurement eliminates the braking phase that occurs when the travel velocity is high. An instrument's validation will never be definitive if it does not have the capacity to measure this expression of velocity.

The significance of the observed differences should have been assessed when analyzing the results with the Bland-Altman plots. If, as mentioned in the manuscript, each 0.07-0.09 m/s represents a change of approximately 5% in the relative intensity (percentages), the magnitude of the differences between the measurements for the same mean (X axis of the graph) can lead to errors close to 15%. This error means that, for the same absolute load, on one occasion, the device indicates that it is, for example, 50% of the 1RM, and on another occasion that it is 65%. Such deviations are unacceptable in a velocity meter. It should be noted that the whole VBT is because each percentage has its velocity, which is very stable (see reference line 461 of the manuscript). If these two premises were not true, the VBT would not exist.

In the manuscript, certain statements do not correspond to the correct velocity control. For example, line 52 states that there is "a strong linear relationship between relative external loads and lifting velocity when a resistance training exercise is performed with maximal concentric effort." For this, some studies are cited, but surprisingly, there are other studies cited in the manuscript that say the opposite and are not cited. The relationship may be linear if there are at least three reasons:
1. The smallest load used in the test is too high.
2. Mean velocity is measured, not MPV.
3. The subjects are used to performing the exercises slowly voluntarily.
In these circumstances, the linear fit is probably the same as the second-degree polynomial fit, but the latter fit is the best when the velocity is correctly measured in a progressive test.

On the other hand, the authors do not present what is the load-velocity adjustment they have found in their study. However, with the loads used in the progressive test, it is impossible to express the participants' maximum performance potential. The number of loads is too few (4-5 loads), and their differences must be lowered. With this test, it is impossible to obtain a correct relationship between absolute or relative loads and their execution velocity.
It does not seem reasonable to state that “the application demonstrated excellent accuracy for measuring RoM during the squat and bench press” (line 282) when significant differences were found: “At the ideal camera position, the 95% LoA were -4.17 to 4.01 cm during the squat and -6.04 to 3.74 cm during the bench press.” (line 294).
If "the average barbell RoM during the squat was 56.00 ± 9.47 cm," it cannot be considered that the full squat exercise was performed. This circumstance is not relevant if it is simply a matter of comparing distances and velocities since what is important is the comparison between them. Still, it does not allow the load-velocity relationship to be established correctly.
The discussion between lines 290 and 307 seems irrelevant. Travel losses (as well as velocity losses) due to fatigue are natural and do not mean an erroneous measurement. What is essential in this case is that the device measures the distance well, whatever it is. Therefore, intra-individual losses are not relevant in this type of analysis.

Experimental design

See basic reporting
No additional comment

Validity of the findings

See basic reporting
No additional comment

Additional comments

No comment

---

## Round 0.2 · accepted · Accept

The reviewers thought the manuscript has been thoroughly corrected and all remarks were addressed by the authors.

·

Basic reporting

Dear authors, I really appreciate your work in case to correct the manuscript and rise the quality of this article. I really appreciate the work which has been done by the other two reviewers where they present their remarks and share their experience.
In my opinion the manuscript has been thoroughly corrected and all my remarks were adressed by the authors.
If this is in accordance with other two reviewers and the editor I recomend publication of this article.

all the best with your future research

Experimental design

response above

Validity of the findings

response above

Additional comments

response above

Reviewer 2 ·

Basic reporting

No comment.

Experimental design

No comment.

Validity of the findings

No comment.

Additional comments

No comment.